# An Attempt to Develop a Model of Brain Waves Using Quantitative Electroencephalography with Closed Eyes in K1 Kickboxing Athletes—Initial Concept

**DOI:** 10.3390/s23084136

**Published:** 2023-04-20

**Authors:** Łukasz Rydzik, Tomasz Pałka, Ewa Sobiło-Rydzik, Łukasz Tota, Dorota Ambroży, Tadeusz Ambroży, Pavel Ruzbarsky, Wojciech Czarny, Marta Kopańska

**Affiliations:** 1Institute of Sports Sciences, University of Physical Education, 31-571 Kraków, Poland; 2Department of Physiology and Biochemistry, Faculty of Physical Education and Sport, University of Physical Education, 31-571 Kraków, Poland; 3Independent Researcher, 35-326 Rzeszów, Poland; 4Department of Sports Kinanthropology, Faculty of Sports, Universtiy of Presov, 08001 Prešov, Slovakia; 5College of Medical Sciences, Institute of Physical Culture Studies, University of Rzeszow, 35-959 Rzeszów, Poland; 6Department of Pathophysiology, Institute of Medical Sciences, Medical College of Rzeszów University, 35-959 Rzeszów, Poland

**Keywords:** brain, QEEG, kickboxing, K-1, brain injury, martial arts

## Abstract

Background: Brain injuries are a common problem in combat sports, especially in disciplines such as kickboxing. Kickboxing is a combat sport that has several variations of competition, with the most contact-oriented fights being carried out under the format of K-1 rules. While these sports require a high level of skill and physical endurance, frequent micro-traumas to the brain can have serious consequences for the health and well-being of athletes. According to studies, combat sports are one of the riskiest sports in terms of brain injuries. Among the sports disciplines with the highest number of brain injuries, boxing, mixed martial arts (MMA), and kickboxing are mentioned. Methods: The study was conducted on a group of 18 K-1 kickboxing athletes who demonstrate a high level of sports performance. The subjects were between the ages 18 and 28. QEEG (quantitative electroencephalogram) is a numeric spectral analysis of the EEG record, where the data is digitally coded and statistically analysed using the Fourier transform algorithm. Each examination of one person lasts about 10 min with closed eyes. The wave amplitude and power for specific frequencies (Delta, Theta, Alpha, Sensorimotor Rhythm (SMR), Beta 1, and Beta2) were analysed using 9 leads. Results: High values were shown in the Alpha frequency for central leads, SMR in the Frontal 4 (F4 lead), Beta 1 in leads F4 and Parietal 3 (P3), and Beta2 in all leads. Conclusions: The high activity of brainwaves such as SMR, Beta and Alpha can have a negative effect on the athletic performance of kickboxing athletes by affecting focus, stress, anxiety, and concentration. Therefore, it is important for athletes to monitor their brainwave activity and use appropriate training strategies to achieve optimal results.

## 1. Introduction

Kickboxing is a combat sport that involves using punches and kicks. In the category of punches, there are several types, including the straight punch, hook punch, uppercut, backfist, and jumping punch. As for kicks, the main techniques include the front, roundhouse, side, downward, and hook kicks, as well as the knee strike. The rules of K-1 allow for the use of all these techniques performed with maximum force in constant combat, which lasts for three, two min rounds with one min intervals, making the fights quite brutal and spectacular. Additionally, the fights take place in a ring, which reduces the escape area and facilitates the direct exchange of strikes between the fighters [1]. Head traumas and brain injuries are a common problem in combat sports, especially in disciplines such as kickboxing. Although, on the one hand, these sports require high skills and physical endurance [2], on the other hand, frequent micro brain injuries can cause serious consequences for the health and well-being of the athletes [3,4]. Head blows, particularly in the temple and forehead areas, can lead to concussion, increased occurrence of post-traumatic stress disorder, and other serious internal injuries [4,5,6,7,8]. Head and brain injuries are becoming an increasingly important topic in numerous scientific studies [9,10,11,12,13,14,15,16]. According to an international panel of experts in the field of sports-related brain injuries, it is important to pay attention to the diagnosis, management, and rehabilitation of brain injuries [15]. Evidence-based guidelines describing the assessment and management of brain injuries in sports have been presented by the American Academy of Neurology [17]. Among the sports disciplines with the highest number of brain injuries, boxing, MMA, and kickboxing are mentioned [18]. In the study by McCrory et al., 2009, 2013, the authors focused on diagnostic protocols and treatment of brain injuries in sports, as well as ways to prevent such injuries. In the research, it has been shown that special attention should be paid to the increased risk of repeated brain injuries in sports which, as a consequence, can lead to serious health consequences [19,20]. Laccarino et al., 2017 focused on mild traumatic brain injuries in combat sports athletes, as well as methods for diagnosing and treating them. In this study, it was indicated that mild traumatic brain injuries in combat sports athletes can lead to serious health problems, such as difficulty concentrating, mood disturbances, and memory-related problems [21]. Bledsoe et al., 2005 conducted a trial on this topic in professional boxing. According to the authors of the study, this risk is very high and represents one of the most serious threats to the health of athletes [22]. In a study conducted by the National Football League (NFL), it was found that 6.1% of NFL players had at least one documented brain injury between 2012 and 2015 [23]. In a study conducted on a group of American football-playing students, it was found that prolonged exposure to repetitive brain injuries can lead to an increased risk of chronic traumatic encephalopathy (CTE) [24].

Research on the brain, conducted using electroencephalography (EEG), is crucial for understanding its workings and how external factors affect its function. However, traditional EEG has limitations in providing detailed information on brain activity in specific areas [25,26]. To overcome these limitations, Quantitative Electroencephalography (QEEG) has been developed as a non-invasive technique for analysing brain electrical activity using scalp electrodes. Compared to traditional EEG, QEEG provides more precise and comprehensive evaluations of brain function, making it useful for identifying subtle changes in brain activity associated with brain dysfunctions, including injuries [27]. EEG measures the brain’s electrical activity, while QEEG provides more detailed information about the characteristics and distribution of this activity in different parts of the brain. QEEG uses advanced mathematical and statistical methods to analyse and interpret EEG data, allowing for the identification of patterns that may indicate various neurological or psychological disorders [28].

Based on the results of quantitative electroencephalography with closed eyes, it can be suggested that the values measured in this manner should be lower. Closing the eyes induces changes in brain activity. When we close our eyes, we significantly reduce the amount of sensory stimuli we receive from the environment, which can lead to a decrease in brain activity in some areas. One of the areas that may undergo changes is the occipital area, which is associated with visual processing. With closed eyes, this area may show lower activity. Additionally, closing the eyes induces a state of rest, which also affects the decrease in brain activity. However, it should be noted that QEEG results with closed eyes are important for assessing the resting state and can provide information on brain activity in such a state [29,30].

In the case of kickboxing athletes, studying brain waves with closed eyes during rest or relaxation can provide valuable insights into their neurological functioning. This is because the brain generates lower-frequency waves during these states, which can reveal abnormalities or deviations that may indicate brain dysfunctions affecting athletic performance or leading to injuries [31]. Furthermore, QEEG can be used to evaluate the impact of training on brain waves, enabling coaches and trainers to optimize training programmes while reducing the risk of injuries [25]. However, studying QEEG in kickboxing athletes presents challenges due to the dynamic and physically demanding nature of the sport. Nevertheless, the information obtained from such studies can be valuable in enhancing athletic performance and reducing potential risks. For instance, identifying brain regions that are particularly active during kickboxing can inform targeted training programmes to improve performance while reducing trauma occurrence. A review of the literature allows to indicate a lack of similar studies, and according to current knowledge, no one has measured K-1 kickboxing fighters using QEEG. The latest QEEG studies mainly concern medical and rehabilitation aspects, but no one has applied this technology to assess athletes practicing striking sports during the break period immediately after the starting period, which can show what changes are induced by performing combat sports.

Therefore, the aim of this study was to develop a model of brain waves using QEEG with closed eyes in K-1 kickboxing athletes. By analysing the data, researchers can gain insights into the neurological functioning of kickboxing athletes and identify any abnormalities or deviations that may indicate brain dysfunctions. Additionally, the study was carried out to assess the impact of training on brain waves and provide information on optimizing training programmes for enhancing performance and reducing the risk of injuries.

## 2. Materials and Methods

The study was conducted among a group of 18 K-1 Rules kickboxing athletes with a high sports level, aged 18 to 28 years. All of the tested athletes were active and competed at the highest level. The sports level was determined on the basis of training experience, the opinion of the leading coach, and sports results. Each tested competitor was a medallist of the highest event in the country, which is the Polish Kickboxing Championships in the K-1 formula organised by the Polish Kickboxing Association.

Detailed selection criteria for the group are presented in Table 1. The study was carried out in accordance with the Declaration of Helsinki and was approved by the Ethics Committee of the University of Rzeszow (protocol code 08/December/2021).

All participants were informed of the study procedures and abstained from participating in sparring fights for 14 days prior to the study. Each participant recorded his dietary intake using the Fitatu smartphone application and was instructed not to consume energy drinks or other stimulants containing caffeine for 48 h before the study. This was aimed at eliminating factors that could distort the test results [32].

### 2.1. QEEG Procedure

The procedure of quantitative electroencephalography (QEEG) involves coding the EEG record digitally and analysing it statistically using the Fourier transform algorithm [33,34]. Each examination of an individual lasted approximately 10 min, with the subject’s eyes closed. The analysis involved measurement of amplitude and power of specific frequencies, while taking the standard norms for adults into account. It was assumed that lower frequency waves had a higher amplitude, with Delta waves being considered normal below 20 µV, Theta below 15 µV, Alpha below 10 µV, the Sensimotor Rhythm (SMR), read Beta 1 and Beta2 below 6 µV, as per the standard. The EEG signal was transformed using the Cz, Pz, and Fz electrodes as the reference site [35] and quantified using Elmiko and DigiTrack software (version 15, PL) (ELMIKO, Warsaw, Poland). Nine channels were recorded for the study, which included Delta, Theta, Alpha, SMR, Beta 1, and Beta2 waves at electrodes (frontal—FzF3F4, central—CzC3C4, and parietal—PzP3P4). The amplitude of QEEG rhythms was calculated using the medical standards of the DigiTrack apparatus. The fast Fourier transform (FFT) algorithm was used to analyse the signal spectrum, with the result of the function being: f(z) = A(z) + j*F(z). The Fourier decomposes the EEG time series into a voltage by frequency spectral graph, commonly called the „power spectrum”, with power being the square of the EEG magnitude, and magnitude being the integral average of the amplitude of the EEG signal, measured from (+) peak-to-(−) peak, across the time sampled [36]. For FFT analysis, the minimum signal amplitude was set to 0.5 µV, with a minimal temporal distance between single maximal values of 0.5 Hz. A computing buffer of 8.2 s (2048 assessment points, accuracy 0.12 Hz) was used for the analysis, resulting in a set of amplitude values for each defined part of the frequency spectrum. The calculation resolution was defined by the gap between single values measured in Hz and was dependent on the signal sampling frequency and the length of the computing buffer, r = fs/N, where r represents the calculation resolution, fs is the signal sampling frequency, and N is the length of the computing buffer. The results of the spectrum analysis in the FFT panel in DigiTrack showed peak-to-peak amplitudes. To ensure appropriate reliability, measurement epochs of several seconds were implemented [31]. The epoch length determined the frequency resolution of the Fourier, with a 1-s epoch providing a 1-Hz resolution (plus/minus 0.5 Hz resolution), and a 4-s epoch providing a 0.25 Hz resolution (plus/minus 0.125 Hz resolution). To eliminate artifacts, the method of blind separation—BSS signals (Blind Source Separation)—were used. This is the estimation of unknown source signals on the basis of registered ones or the extraction of interfering signals for their subsequent elimination. The source of artifacts, i.e., undesirable components in the EEG signal, may be: heart rate, eye movement (blinking), facial expressions (facial muscle movements), jaw movement or swallowing, and chest movement during breathing. In addition, the value of the signal measured outside the brain activity at the measurement site consists of signals from other areas of the brain and other disturbances from outside the body. Blind signal separation algorithms in EEG-related research are aimed at removing the aforementioned artifacts as precisely as possible, so that in the further steps of signal analysis only process those from the areas of the cerebral cortex of interest [27].

### 2.2. Statistical Analysis

Statistical analysis of the collected data was conducted using Statistica v.13.3 (TIBCO Software, Palo Alto, Santa Clara, CA, USA) software. Basic descriptive statistics were calculated, including mean, standard deviation, lower and upper quartiles, minimum and maximum values. The significance of differences between individual channels was calculated using one-way ANOVA for dependent groups, with a significance level set at *p* < 0.05. The choice of test was based on the assumption of normal distribution, which was verified using the Shapiro-Wilk test, and homogeneity of variance, which was verified using the Levene’s test. Additionally, effect size was reported based on eta-squared values. The results of the study were presented in graphs created using Canva, and the normative scale for QEEG values was adopted as follows: Delta—up to 20 µV, Theta—up to 15 µV, Alpha—up to 10 µV, SMR—6 µV, Beta I—6 µV, Beta2—6 µV [37,38].

## 3. Results

It should be noted that the average Delta values are diverse in different brain regions, suggesting that Delta activity is more concentrated in the frontal area than in others. Additionally, the minimum and maximum values and quartiles suggest that in some areas, Delta activity is more variable than in others. The average values are within the reference range, but the maximum values significantly exceed it. High maximum values were recorded in all leads. In the frontal lobe, the variation in leads appeared statistically significant (Table 2, Figure 1).

The highest average Theta activity was observed in Fz, while the lowest average activity was noted in P4. Standard deviations also differed across different electrode locations, with Fz having the highest, indicating greater variability in Theta activity for this region. The average values were within the reference range. However, the recorded maximum values in the FZ, F3, and F4 leads were outside the reference range (Table 3, Figure 2).

In the assessment of the Alpha wave, the highest peak amplitude was observed at C3, while the lowest peak was observed at F3. The standard deviation values indicate that the greatest variability in Alpha activity was observed at the peak amplitudes of the skull (Cz) and parietal regions (C3 and C4), which exceeded the reference norms. It is worth noting the asymmetry observed between F3 and F4, which exceeded a difference of 20% (Table 4, Figure 3).

In Table 5, statistical characteristics of the SMR amplitude parameters are presented. The highest amplitude value was observed in F4, while the lowest, in C4. The largest dispersion degree of values from the mean was noted for the lateral frontal electrodes (F3; F4). The value of F4 exceeded the reference standard (Figure 4).

In the Beta frequency range, the highest activity was observed in the right frontal lobe (F4), while the lowest was in its central part (Fz). The standard deviation values show that the greatest variability in activity was noted in the case of F4 amplitude. Values for F3, F4, and P3 exceeded the reference norms (Table 6, Figure 5).

In Table 7, the characteristics of Beta2 waves are presented. The highest activity was observed in the frontal lobes of both the left and right hemispheres, while the lowest was noted in the central frontal lobe. The values for all electrodes exceeded the reference range (Figure 6).

## 4. Discussion

QEEG (Quantitative Electroencephalography) with eyes closed is a type of neuroimaging technique used to measure and analyse electrical activity of the brain. When a person closes their eyes during a QEEG recording, it can show the brain’s resting-state activity, also known as the alpha rhythm.

The findings of our own study allow us to indicate that the average values for the Delta and Theta frequencies did not exceed the accepted reference norm. Delta waves are associated with regenerative processes and deep sleep, which are crucial for high-performance athletes [39,40,41]. Therefore, appropriate values of Delta waves may indicate the proper regeneration processes necessary in sports [42]. On the other hand, the regularity of Theta waves may indicate good concentration, which is essential for preparing for a fight, especially within a psychological context [43]. These results highlight the importance of monitoring brain activity using QEEG in sports performance evaluations. It can provide valuable insights into the athlete’s cognitive and psychological states, which are crucial factors in high-performance sports [44]. Further research can expand on the significance of QEEG in sports and the potential benefits of using this technology to improve athletes’ performance and well-being.

In the present study, high values were recorded for Alpha waves in CZ, C3, and C4 leads, which exceeded the reference norm. The results of the present study suggest that high values were recorded for Alpha waves in certain leads, indicating a state of relaxation with closed eyes. This is in line with previous research in which it has been shown that the amplitude of Alpha waves usually increases during relaxation, which is characterised by a decrease in visual activity and a greater focus on one’s own thoughts [45,46]. Alpha waves appear in the temporal region on the right side of the brain. Their level increases not only during creative tasks but also in a relaxed state. They stimulate dreams and promote inspiration. Sharpened alpha waves in the occipital region indicate deep relaxation. Alpha waves are also present in the frontal lobe. When they are present and repetitive, they can cause anxiety, indicating that the patient is under tension, perhaps before facing a challenge. Too low levels of alpha waves can cause sleep problems, generate fear, and anxiety. Conversely, when alpha waves are too high, we have difficulties concentrating and feel a lack of energy. Numerous studies report that alpha rhythm largely depends on age and gender [45]. However, it is important to consider that individual differences may affect the results, such as age, gender, health status, medication, stress, and fatigue [47,48,49]. During sports competition or training, athletes are exposed to a high level of stress and physical exertion, leading to muscle tension and emotional tension. When an athlete closes his/her eyes, it can help relax and reduce muscle tension, which can lead to an increase of Alpha waves in the brain. This can be beneficial for athletes, as relaxation and reduced muscle tension can lead to better performance and improved overall well-being [50]. Higher levels of the alpha band improve heart rate variability, which is associated with lower levels of anxiety and stress [51]. It is worth noting that Alpha wave activity can also vary depending on the athlete’s level of experience. In a study on experienced karate athletes, lower levels of Alpha wave activity were associated with greater efficiency in processing sensory information. This suggests that experienced athletes may have developed a more efficient neural network for processing sensory information, leading to a different pattern of brain activity during competition or training [52].

In the analysis carried out in the current study, an increase above the reference norm in the sensomotoric SMR waves in electrode f4 has been shown. Sensomotoric rhythm (SMR) is a type of brainwave that occurs within the frequency range of 12–15 Hz. It is primarily found in the sensory and motor areas of the brain and is associated with the activation of the motor cortex. SMR waves are used in biofeedback training as a measure of relaxation and concentration levels, and athletes can use biofeedback training to improve their physical and mental performance. The biofeedback training process involves the athlete learning to control their physiological responses, such as heart rate, breathing, and brainwave activity, achieving a desired state of relaxation or concentration [53,54]. Therefore, increased SMR may indicate that the athletes were in a state of full relaxation.

Budde et al., 2008 have proven that coordination training increases SMR waves, which may improve attention and concentration function [55]. Coordination training involves practicing movements that require the synchronisation of different body parts and can improve a person’s ability to focus attention on specific tasks, such as performing a complex kickboxing combination. Therefore, it is possible that the elevated SMR waves observed in the kickboxing athletes in this study were due to the structure of their training, which includes significant focus on coordination exercises combining foot techniques with hand techniques [56].

In the Beta 1 frequency range, an observation was made that indicated an increase in activity in the frontal leads F3 and F4. This finding could potentially suggest the presence of excessive activity in the prefrontal cortex, a region of the brain that is associated with executive functions, decision-making and emotion regulation. An overactive prefrontal cortex has been linked to various negative psychological states, such as stress and anxiety, which may manifest themselves in physical symptoms such as headaches, muscle tension and fatigue. Furthermore, such hyperactivity in the prefrontal cortex may lead to difficulties in decision-making, as well as impairments in other cognitive processes such as working memory and attention. Therefore, it is important to further investigate this finding and explore potential interventions or strategies to help regulate prefrontal cortex activity and alleviate related symptoms [57]. The increase of Beta 1 waves in the frontal lobe may be a consequence of numerous blows received in the head area, which are specific to kickboxing in the K-1 formula [58]. In scientific research, it has been confirmed that people who have suffered a brain injury exhibit increased beta wave activity in the frontal lobe compared to healthy individuals [59,60].

In the Beta2 frequency, all average values were outside the reference range for all leads. This may suggest intense brain activity. Scientists have repeatedly verified elevated Beta2 waves in sports and have indicated that they may be related to sports performance [61]. The fact that high values were recorded with closed eyes in all leads is intriguing. This study indicates the need for further analysis and verification of brain changes among kickboxing athletes, as well as a detailed examination as to whether the numerous blows received during training and competition have negative impact on the athletes’ health.

The study conducted on kickboxing athletes competing in the K-1 formula revealed elevated values in several brain waves when measured with closed eyes. The increased Alpha waves observed in the athletes could be attributed to the relaxation and resting state that occurs when the eyes are closed. However, the elevated values in SMR, Beta 1, and Beta2 waves indicate that the athletes were in a focused and alert state, potentially due to the anticipation and stress associated with competition. These findings suggest that K-1 kickboxing may have impact on the brain state of athletes, particularly during the competition period. The elevated SMR waves could indicate a state of focused attention and readiness for action, as this frequency band is associated with motor planning and execution. Meanwhile, the increased Beta 1 and Beta2 waves could indicate a state of mild stress or anxiety, as these frequencies are associated with cognitive and emotional processing, particularly related to decision-making and problem-solving. The study results suggest that further research is needed to better understand the impact of K-1 kickboxing on the brain state of athletes, particularly during the competition period.

### Limitations of the Study

The main limitation of this study is the relatively small number of K-1 formula kickboxing athletes. Additionally, quantitative electroencephalography was performed with only nine leads of the brain cortex. Expanding the study to a full-head QEEG measurement would be worthwhile, although in this research, we selected the most important areas of the brain cortex.

## 5. Conclusions

The conclusions of our research regarding kickboxers indicate that too high activity of SMR, Beta 1 and Beta2 waves, as well as too high and long-lasting activity of Alpha waves, can negatively affect the achievement of optimal sports performance. This is also unfavourable for humans because functioning is carried out at the expense of other more desirable brain functions. Excessive activity of these waves can lead to feelings of excessive focus, stress, anxiety, difficulty concentrating, reduced motor reactions and impaired alertness. Consequently, in sports, including kickboxing, too high activity of these waves can have a negative impact on success in a competition.

### Practical Implications

On the basis of our results, it may be concluded that in sports, including kickboxing, it is important to monitor and control the brainwave activity of athletes. Due to this, excessive focus, anxiety, stress, and difficulties with maintaining attention and concentration can be prevented, which could otherwise have a potentially negative effect on the competition results. The use of appropriate techniques and methods, such as meditation training, biofeedback, or neurofeedback, can help regulate brain wave activity and improve athletic performance. Therefore, our findings may have practical applications in sports training and preparation, including kickboxing, helping athletes achieve better performance and success in competitions.

## Figures and Tables

**Figure 1 sensors-23-04136-f001:**
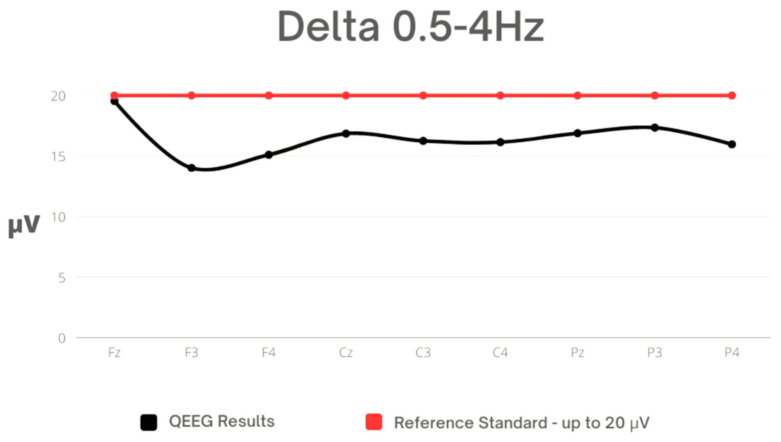
Mean values for Delta against the reference standard.

**Figure 2 sensors-23-04136-f002:**
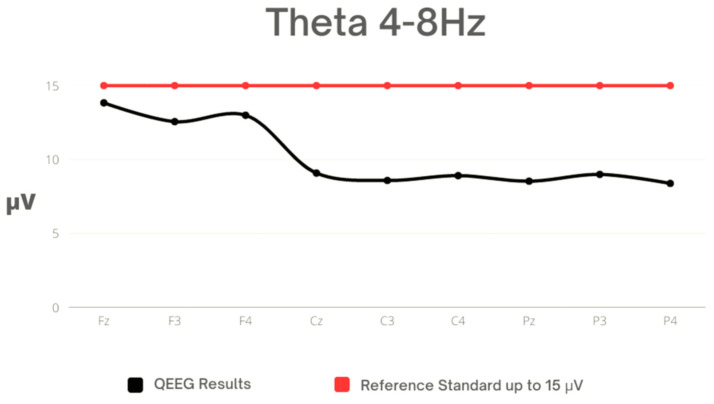
Mean values for Theta against reference standard.

**Figure 3 sensors-23-04136-f003:**
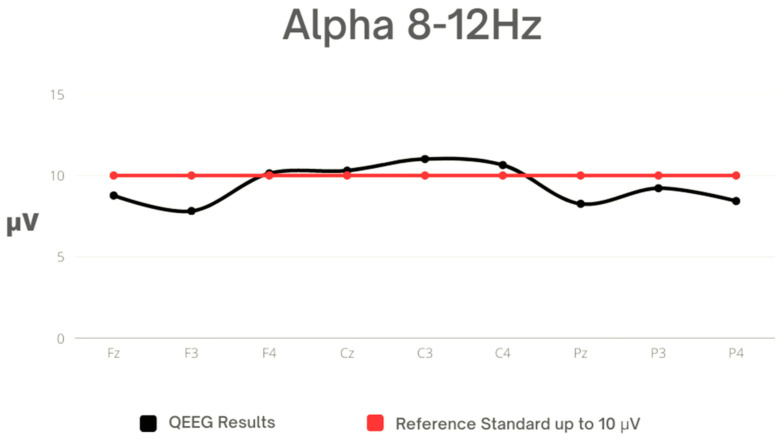
Mean values for Alpha against reference standard.

**Figure 4 sensors-23-04136-f004:**
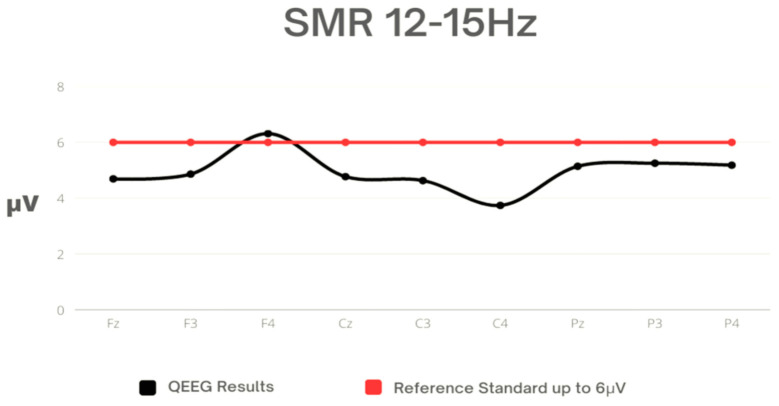
Mean values for SMR against the reference standard.

**Figure 5 sensors-23-04136-f005:**
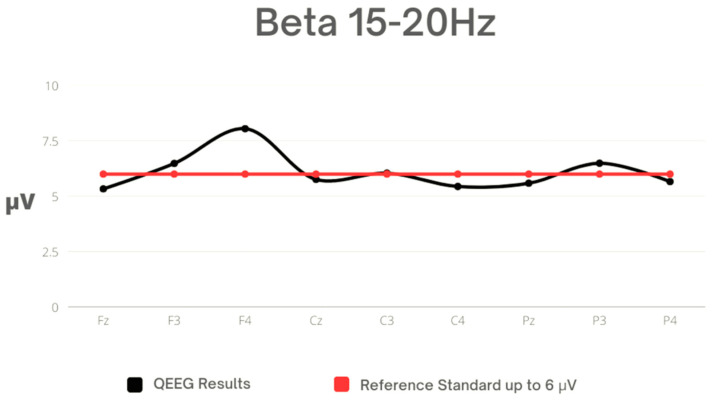
Mean values for Beta 1 against the reference standard.

**Figure 6 sensors-23-04136-f006:**
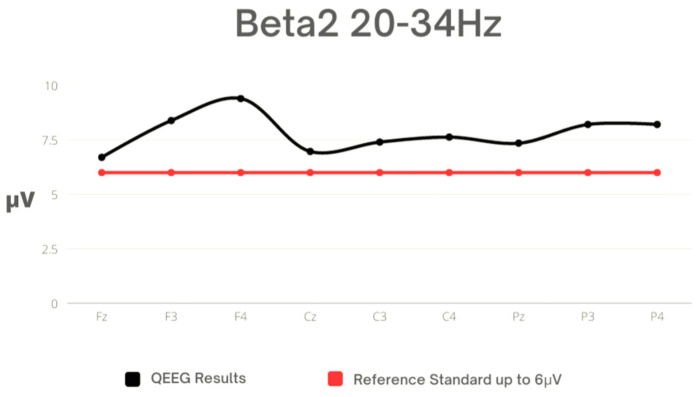
Mean values for Beta2 against the reference standard.

**Table 1 sensors-23-04136-t001:** Inclusion and exclusion criteria.

Inclusion Criteria	Exclusion Criteria
Training experience of at least 10 years	Neurological disorders
Current medical examinations	Skin diseases of the head
Participation in at least 5 competitions per year (having medal from national championship)	Cranio-Cerebral Injuries
	Taking psychotropic drugs
	Injuries and severe knockouts

**Table 2 sensors-23-04136-t002:** Descriptive statistics of Delta waves from all channels (µV).

Delta 0.5–4 Hz	Mean	Min	Max	Q1	Q3	SD
Fz	19.54	13.03	33.78	13.84	22.19	7.50
F3	14.01	11.48	22.84	12.19	14.21	3.01
F4	15.10	10.62	26.35	12.28	16.00	4.33
	*p* < 0.001 ES = 0.23					
Cz	16.85	9.13	24.98	13.03	19.45	4.47
C3	16.24	12.19	23.23	13.98	19.43	3.54
C4	16.14	10.62	26.35	13.28	18.62	3.87
	*p* = 0.86 ES = 0.00					
Pz	16.88	10.62	26.35	12.28	22.28	5.32
P3	17.34	11.48	34.21	12.83	21.99	6.78
P4	15.96	10.62	26.35	13.28	19.45	4.13
	*p* = 0.61 ES = 0.03					

**Table 3 sensors-23-04136-t003:** Descriptive statistics for Theta waves from all channels (µV).

Theta 4–8 Hz	Mean	Min	Max	Q1	Q3	SD
Fz	13.83	9.19	27.55	11.77	14.91	4.20
F3	12.56	9.88	17.19	11.78	13.70	1.86
F4	12.99	9.90	18.37	11.91	14.37	2.14
	*p* = 0.34 ES = 0.06					
Cz	9.08	6.69	11.67	7.38	10.10	1.66
C3	8.58	6.59	11.10	7.63	10.59	1.69
C4	8.91	5.78	11.78	6.71	11.78	2.48
	*p* = 0.51 ES = 0.04					
Pz	8.53	5.14	10.62	6.85	10.62	2.11
P3	8.99	6.01	10.93	7.54	10.12	1.68
P4	8.38	5.08	10.08	7.25	10.08	1.85
	*p* = 0.36 ES = 0.06					

**Table 4 sensors-23-04136-t004:** Descriptive statistics of Alpha waves from all channels (µV).

Alpha 8–12 Hz	Mean	Min	Max	Q1	Q3	SD
Fz	8.77	6.31	14.34	6.87	9.31	2.77
F3	7.81	5.45	11.56	6.77	7.58	1.89
F4	10.13	5.58	35.37	6.11	10.12	7.33
	*p* = 0.27 ES = 0.08					
Cz	10.30	5.58	18.58	7.53	10.70	4.17
C3	11.02	5.34	19.79	8.34	16.71	4.42
C4	10.64	7.01	19.88	7.14	16.18	4.40
	*p* = 0.71 ES = 0.03					
Pz	8.26	4.69	12.07	6.18	9.94	2.32
P3	9.22	4.68	13.09	7.68	11.69	2.62
P4	8.43	4.60	11.98	7.08	9.94	2.33
	*p* = 0.21 ES = 0.08					

**Table 5 sensors-23-04136-t005:** Descriptive statistics of SMR waves from all channels (µV).

SMR 12–15 Hz	Mean	Min	Max	Q1	Q3	SD
Fz	4.69	3.74	6.14	4.00	5.08	0.81
F3	4.86	0.87	3.67	6.11	3.83	5.31
F4	6.31	4.37	3.12	15.64	3.58	5.25
	*p* = 0.60 ES = 0.03					
Cz	4.77	3.02	7.02	3.68	5.39	1.11
C3	4.63	4.21	7.58	4.27	4.59	0.76
C4	3.74	3.21	5.41	3.45	3.92	0.49
	*p* = 0.002 ES = 0.29					
Pz	5.14	3.47	7.38	4.18	5.76	1.28
P3	5.25	3.47	8.80	4.39	5.80	1.45
P4	5.18	3.18	9.18	3.58	5.66	1.76
	*p* = 0.97 ES = 0.001					

**Table 6 sensors-23-04136-t006:** Descriptive statistics for Beta waves from all channels (µV).

Beta 15–20 Hz	Mean	Min	Max	Q1	Q3	SD
Fz	5.33	4.50	6.59	5.03	5.63	0.67
F3	6.48	4.98	9.77	5.60	6.77	1.61
F4	8.05	4.09	19.30	5.58	6.69	5.25
	*p* = 0.02 ES = 0.20					
Cz	5.76	4.81	7.59	5.20	6.19	0.94
C3	6.04	4.80	7.83	5.38	6.93	1.06
C4	5.44	4.21	6.88	4.82	6.27	0.92
	*p* < 0.001 ES = 0.61					
Pz	5.59	4.21	7.51	4.44	6.35	1.06
P3	6.49	4.91	7.96	5.89	7.44	1.04
P4	5.66	4.43	7.51	4.44	6.35	1.11
	*p* = 0.01 ES = 0.22					

**Table 7 sensors-23-04136-t007:** Descriptive statistics for Beta2 waves across all channels (µV).

Beta2 20–35 Hz	Mean	Min	Max	Q1	Q3	SD
Fz	6.70	5.66	7.96	6.16	7.31	0.77
F3	8.39	6.06	14.67	7.31	9.41	2.01
F4	9.40	5.21	15.27	7.08	11.17	2.70
	*p* < 0.001 ES = 0.43					
Cz	6.97	5.18	9.25	6.25	7.49	0.86
C3	7.40	6.14	9.82	6.40	7.64	1.00
C4	7.63	6.38	9.28	6.69	7.89	0.86
	*p* = 0.04 ES = 0.17					
Pz	7.35	6.18	9.28	6.69	7.81	0.83
P3	8.21	6.79	9.29	7.46	8.79	0.80
P4	8.21	5.85	9.56	7.15	8.79	0.94
	*p* = 0.002 ES = 0.31					

## Data Availability

All data was included in the manuscript.

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
