# Peer review of "An Attempt to Develop a Model of Brain Waves Using Quantitative Electroencephalography with Closed Eyes in K1 Kickboxing Athletes—Initial Concept"

_sensors, 2023, doi:10.3390/s23084136_

Round 1

Reviewer 1 Report

“Attempt to develop a model of brain waves using quantitative electroencephalography with closed eyes in K1 kickboxing athletes.”

Overall strengths of the article:

This manuscript aims to develop a model of brain waves using quantitative electroencephalography (QEEG) with closed eyes on kickboxing athletes competing in the K1 formula. This study was conducted on a group of 18 K1 kickboxing athletes, 18-28 years old, demonstrating a high level of sports performance. The Delta, Theta, Alpha, SMR, Beta-1, and Beta-2 wave amplitude and power were analyzed in nine leads. High values were shown in the Alpha frequency in central leads, SMR in lead F4, Beta 1 in leads F4 and P3, and Beta 2 in all leads. The increased Alpha waves observed in the athletes could be attributed to the relaxation and resting state that occurs when the eyes are closed. However, the elevated values in SMR, Beta 1, and Beta 2 waves indicate that the athletes were in a focused and alert state, potentially due to the anticipation and stress associated with the competition. These findings suggest that K1 kickboxing may have an impact on the brain state of athletes, particularly during the competition period. The elevated SMR waves could indicate a state of focused attention and readiness for action, as this frequency band is associated with motor planning and execution. Meanwhile, the increased Beta 1 and Beta 2 waves could indicate a state of mild stress or anxiety, as these frequencies are associated with cognitive and emotional processing, particularly related to decision-making and problem-solving. The study's results suggest that further research is needed to better understand the impact of K1 kickboxing on the brain state of athletes, particularly during the competition period.

There is much missing information in the presentation of the methods and results, which substantially impede understanding and clarity. In the current state of the manuscript, it gives the impression that the results are unreliable, and may not efficiently convey and support the message presented by the Authors. Details are in the specific comments section.

Specific comments on weaknesses:

Major concerns:

1.     The results of this manuscript are based on a small sample size (n=18). This limits their strength and impact. Further, this issue is compounded by analyzing only the amplitude and power of specific frequencies. It was pre-assumed that the amplitude of the waves decreased as the frequency of the waves decreased. Only 9 leads on the brain cortex, a single measurement for only 10 minutes even not a full-head QEEG measurement. In my eyes, the presented results do not allow us to go beyond to make any significant conclusion.

2.     Methods need to be presented in a better way, in the present form it is not very clear without any traces of recording of waveforms.

3.     Results: I have a serious concern about the way results were presented as the minimum and maximum values without units have no meaning. It is not clear what they have presented in all these tables. Similarly, in the figure, the mean values have no meaning without any unit. What are the reference standards? I strongly suggest rewriting the result section with complete information.

4.     What is the reference standard? It is not described anywhere in the manuscript why? It is hard to make any conclusion from the results that have been presented here.

5.     Confusing discussion, it is not clear what is new and what is already known. Similarly, the conclusion, in its present form it seems this study has no value.

Minor points:

1.     Abstract: Please avoid abbreviations in the abstract (e.g., MMA, SRM, f4, p3, etc.) or explain them.

2.     What are the FFT algorithm and FFT analysis not very clear from the method presented.

Author Response

Dear Reviewer,

Thank you very much for your time and valuable comments, which all have been considered and incorporated. The detailed list of responses is given below. We hope that the modifications and explanation will be acceptable for you.

Yours sincerely,

Rydzik, corresponding author

There is much missing information in the presentation of the methods and results, which substantially impede understanding and clarity. In the current state of the manuscript, it gives the impression that the results are unreliable, and may not efficiently convey and support the message presented by the Authors. Details are in the specific comments section. There is much missing information in the presentation of the methods and results, which substantially impede understanding and clarity. In the current state of the manuscript, it gives the impression that the results are unreliable, and may not efficiently convey and support the message presented by the Authors. Details are in the specific comments section.

A: Thank you for your review, all comments have been taken into account. Additionally, the entire manuscript has been reviewed and corrected by a Native Speaker and certified for language quality.

Major concerns:

  1. The results of this manuscript are based on a small sample size (n=18). This limits their strength and impact. Further, this issue is compounded by analyzing only the amplitude and power of specific frequencies. It was pre-assumed that the amplitude of the waves decreased as the frequency of the waves decreased. Only 9 leads on the brain cortex, a single measurement for only 10 minutes even not a full-head QEEG measurement. In my eyes, the presented results do not allow us to go beyond to make any significant conclusion.

A: Thank you very much for your comment and thorough review. Kickboxing is a combat sport that distinguishes between different forms of division, including board and ring styles. In board styles, fighters compete under rules such as kicklight, lightcontact, and point-fighting, while ring styles include full contact (which prohibits kicks below the belt and precise strikes after turning), low kick (which prohibits precise techniques), and K1 (which allows all techniques). It is a distinct discipline in which fighters specialize in basic fights, and as a result, the population is limited. Under the link, we present all the students for kickboxing K1 performances in Poland, which are the pro-am championships (a competition that brings together the highest-level fighters). We note that there are 47 male fighters, and we would like to display 18 high-quality levels as an attractive sample. In addition, the sample size was confirmed by G*power.

http://pzkickboxing.pl/soi/zawodnicy/226

Regarding the 9 measurements, such studies have already been conducted on patients, so we considered this procedure to be the most appropriate. The head injuries that fighters receive are directed at the frontal, temporal, and occasionally occipital lobes, so the most vulnerable points were examined. We conducted only one study because we wanted to examine the fighters' state outside the fight (when they are not in training), which shows us what happens to them as a result of long-term participation in the sport. However, we agree with your comment and have modified the title to show that it is a preliminary concept. A detailed review of the literature indicates that no one has conducted such studies yet, so we believe that the novelty aspect is attractive. We hope you will give us a chance.

  1. Methods need to be presented in a better way, in the present form it is not very clear without any traces of recording of waveforms.

A: The methods have been rewritten and corrected. Thank you for your comment.

  1. Results: I have a serious concern about the way results were presented as the minimum and maximum values without units have no meaning. It is not clear what they have presented in all these tables. Similarly, in the figure, the mean values have no meaning without any unit. What are the reference standards? I strongly suggest rewriting the result section with complete information.

A: Thank you for your comment, it's very valuable. We have added units to both the tables and the graphs.

  1. What is the reference standard? It is not described anywhere in the manuscript why? It is hard to make any conclusion from the results that have been presented here.

            A: This information has been described in the QEEG Procedure and the statistical analysis method. Additionally, references have been provided for the source of the reference values.

  1. Confusing discussion, it is not clear what is new and what is already known. Similarly, the conclusion, in its present form it seems this study has no value.

A: Thank you for your comment, we have revised the discussion and rewritten the conclusions.

Minor points:

  1. Abstract: Please avoid abbreviations in the abstract (e.g., MMA, SRM, f4, p3, etc.) or explain them.

A: This has been corrected

  1. What are the FFT algorithm and FFT analysis not very clear from the method presented.

A: This has been corrected. Dodano informację w sekcji QEEG Procedure

Reviewer 2 Report

Dear authors,

After reading my comments, do not feel discouraged. 

Abstract:

“Conclusions: Kickboxing athletes competing in the K1 formula demonstrate elevated values in the study with closed eyes in the Alpha, SMR, Beta 1, and Beta 2 waves. Elevated Alpha waves are associated with closing the eyes. Elevated SMR, Beta 1, and Beta 2 waves may indicate focus on action and mild stress, especially during the competition period. The results of the study indicate the need for more detailed analyses and determining the im-pact of K1 kickboxing on the brain state of athletes.”

 This conclusion is confusing. The fact that you are explaining “Elevated Alpha waves are associated with closing the eyes. Elevated SMR, Beta 1, and Beta 2 waves may indicate focus on action and mild stress, especially during the competition period.” in the conclusion, is non-relevant. You can add that information in the intro or somewhere else. Moreover, we want to know what it means.

Introduction:

What is K1 and what are the rules of K1 that are so brutal?

“According to a study, combat sports are among the most risky sports in terms of brain injuries.” – You are repeating this information so many times… we already understood it from the first time you said it and cited it in the literature.

When you say, for example, ‘McCrory et al.”, you need to add the year. And you do that many times. Please, correct this.

“Brain research using electroencephalography (EEG) is crucial for understanding the workings of the brain and how external factors affect its function. However, traditional EEG has limitations in providing detailed information on brain activity in specific areas 25].” – What specific areas are we talking about? Because QEEG is literally EEG, so it measures the same. And saying that QEEG can solve the limitation found by using EEG is really confusing. You need to better explain the method here so that readers can understand the difference and why QEEG can be a useful tool for what you want to explore.

“In the case of kickboxing athletes, studying brain waves with closed eyes during rest or relaxation can provide valuable insights into their neurological functioning. This is because the brain generates lower frequency waves during these states, which can reveal abnormalities or deviations that may indicate brain dysfunctions affecting athletic performance or lead to injuries.” – These are brave sentences without reference. If you wrote that, at least show us some strong references about that.

“The latest QEEG studies mainly concern medical and rehabilitation aspects, but no one has applied this technology to assess athletes practicing striking sports.” – Neither do you... I am only in the Introduction, but if you measure QEEG with eyes closed, I am assuming that they are in the resting state. “Not practicing”… I believe that it is a problem with the English. This is completely natural if you are not a native, but you must reformulate, otherwise, the readers will not understand what you really mean.

Materials and methods:

What do you mean by ‘high sports level’? Are they elite, medals won, what exactly is high sport level for you? Please add that information.

I don’t think it is a major problem; however, it would take 2 minutes to build a better table without that many lines.

You should recheck what the ‘inclusion and exclusion criteria’ are... Furthermore, if you say ‘Training experience of at least 10 years’, you don’t really need to say ‘Training experience less than 10 years’. It is redundant…

Why have they abstained from sparing the previous 14 days, where is the reference for that? The same applies to the 48 hours of caffeine.

“It was assumed that the amplitude of the waves decreased as the frequency of the waves decreased” – This is wrong… lower frequency waves have higher amplitude… vice versa.

“The elimination of artifacts from the EEG recording was performed both manually and automatically [26].” – This is one of the most demanding steps. Need more explanation on how you did it, especially the manual part.

I am afraid I can move forward with the revision, since I detected so many major problems that already need to be solved.

Author Response

Dear Reviewer,

Thank you very much for your time and valuable comments, which all have been considered and incorporated. The detailed list of responses is given below. We hope that the modifications and explanation will be acceptable for you.

Yours sincerely,

Rydzik, corresponding author

After reading my comments, do not feel discouraged. 

A: Thank you

Abstract:

“Conclusions: Kickboxing athletes competing in the K1 formula demonstrate elevated values in the study with closed eyes in the Alpha, SMR, Beta 1, and Beta 2 waves. Elevated Alpha waves are associated with closing the eyes. Elevated SMR, Beta 1, and Beta 2 waves may indicate focus on action and mild stress, especially during the competition period. The results of the study indicate the need for more detailed analyses and determining the im-pact of K1 kickboxing on the brain state of athletes.”

 This conclusion is confusing. The fact that you are explaining “Elevated Alpha waves are associated with closing the eyes. Elevated SMR, Beta 1, and Beta 2 waves may indicate focus on action and mild stress, especially during the competition period.” in the conclusion, is non-relevant. You can add that information in the intro or somewhere else. Moreover, we want to know what it means.

 A: The conclusions have been rewritten, thank you for your comment

Introduction:

What is K1 and what are the rules of K1 that are so brutal?

A:More information added

“According to a study, combat sports are among the most risky sports in terms of brain injuries.” – You are repeating this information so many times… we already understood it from the first time you said it and cited it in the literature.

A: Irrelevant repetitions have been deleted, the remaining uses paraphrased.

When you say, for example, ‘McCrory et al.”, you need to add the year. And you do that many times. Please, correct this.

A: This has been corrected

“Brain research using electroencephalography (EEG) is crucial for understanding the workings of the brain and how external factors affect its function. However, traditional EEG has limitations in providing detailed information on brain activity in specific areas 25].” – What specific areas are we talking about? Because QEEG is literally EEG, so it measures the same. And saying that QEEG can solve the limitation found by using EEG is really confusing. You need to better explain the method here so that readers can understand the difference and why QEEG can be a useful tool for what you want to explore.

A: This has been corrected

“In the case of kickboxing athletes, studying brain waves with closed eyes during rest or relaxation can provide valuable insights into their neurological functioning. This is because the brain generates lower frequency waves during these states, which can reveal abnormalities or deviations that may indicate brain dysfunctions affecting athletic performance or lead to injuries.” – These are brave sentences without reference. If you wrote that, at least show us some strong references about that.

A: Added references  

“The latest QEEG studies mainly concern medical and rehabilitation aspects, but no one has applied this technology to assess athletes practicing striking sports.” – Neither do you... I am only in the Introduction, but if you measure QEEG with eyes closed, I am assuming that they are in the resting state. “Not practicing”… I believe that it is a problem with the English. This is completely natural if you are not a native, but you must reformulate, otherwise, the readers will not understand what you really mean.

A: This has been corrected

Materials and methods:

What do you mean by ‘high sports level’? Are they elite, medals won, what exactly is high sport level for you? Please add that information.

A: This has been corrected

I don’t think it is a major problem; however, it would take 2 minutes to build a better table without that many lines.

A: This has been corrected

You should recheck what the ‘inclusion and exclusion criteria’ are... Furthermore, if you say ‘Training experience of at least 10 years’, you don’t really need to say ‘Training experience less than 10 years’. It is redundant…

A: This has been corrected

Why have they abstained from sparing the previous 14 days, where is the reference for that? The same applies to the 48 hours of caffeine.

A: Added more information  

“It was assumed that the amplitude of the waves decreased as the frequency of the waves decreased” – This is wrong… lower frequency waves have higher amplitude… vice versa.

A: Thank you. This has been corrected

“The elimination of artifacts from the EEG recording was performed both manually and automatically [26].” – This is one of the most demanding steps. Need more explanation on how you did it, especially the manual part.

 A: Added more information. This has been corrected

I am afraid I can move forward with the revision, since I detected so many major problems that already need to be solved.

A: We hope for the changes to be acceptable to you. In addition, the work has been revised by a Native Speaker. Thank you for your valuable suggestions

Round 2

Reviewer 1 Report

In the revised manuscript authors have successfully addressed all the comments raised by the reviewer and incorporated all the suggestions to improve the quality of the paper. I think this manuscript has been sufficiently improved from the previous version.

Author Response

Dear Reviewer, 
Thank you for your valuable time. Your comments have helped us a lot and for that we especially thank you. In addition, our manuscript has been improved again. 

Your Sincerly, 

Łukasz Rydzik 

Reviewer 2 Report

Dear authors,

When answering a reviewer, please try to always write the modifications in the answer. Otherwise, reviewers must check all the documents again to remember what they asked for and check if it was or not truly corrected… we have a few days to answer, and your paper is not the only work we have.

 Methods:

You still do not understand my point. If your Inclusion criteria are “Current medical examination”, obviously, we know that participants without current medical examination are not allowed. You do not need to write it down. Is the same as saying that you are turning LEFT… obviously if you are turning LEFT, then you are NOT turning RIGHT. I hope you understood my point. Moreover, check this paper to understand the difference between INCLUSION vs. EXCLUSION criteria: https://www.ncbi.nlm.nih.gov/pmc/articles/PMC6044655/

What you did in Table 1 with taking the lines is highly recommended for the other tables. Check out other papers. But again, it is not a major concern, but I believe that the MDPI editor will ask you for that…

And, probably reviewer 2 asked for it, you do not need to place the units in each value. When you quote the table you can say that the values presented are in microvolts. Easier and faster (now that you had all that work, let it be).

Results and Discussion:

In the figure, you wrote Alfa, while Alpha.

I do not get the point of assessing Alpha in the frontal area… alpha is mainly in the occipital area, so for this work I do not understand. Furthermore, “The results of the present study suggest that high values were recorded for Alpha waves in certain leads, indicating a state of relaxation with closed eyes. This is in line with previous research in which it has been shown that the amplitude of Alpha waves usually increases during relaxation, which is characterized by a decrease in visual activity and a greater focus on one's own thoughts.” – You lack the reference(s) and you cannot say that your results are in line with the literature if the other studies do not show higher levels at the same sites as your study. This is a completely different story. If you find a study with the same results, on the same sites, then your results are in line with the literature.

To improve your discussion about alpha, when you talk about relaxation, you can mention a recent article on how higher levels of the alpha band improve heart rate variability associated with lower levels of anxiety and stress (Domingos et al. 2021)*

* Domingos, C., Silva, C. M. D., Antunes, A., Prazeres, P., Esteves, I., & Rosa, A. C. (2021). The influence of an alpha band neurofeedback training in heart rate variability in athletes. International Journal of Environmental Research and Public Health, 18(23), 12579.

Overall, I do hope that the results you had were well cleaned of artifacts and others… in future works, I would recommend working with a neuroscientist with experience in Matlab and coding.

Author Response

Dear Reviewer,

Thank you very much for your time and valuable comments, which all have been considered and incorporated. The detailed list of responses is given below. We hope that the modifications and explanation will be acceptable for you.

Yours sincerely,

Rydzik, corresponding author

You still do not understand my point. If your Inclusion criteria are “Current medical examination”, obviously, we know that participants without current medical examination are not allowed. You do not need to write it down. Is the same as saying that you are turning LEFT… obviously if you are turning LEFT, then you are NOT turning RIGHT. I hope you understood my point. Moreover, check this paper to understand the difference between INCLUSION vs. EXCLUSION criteria: https://www.ncbi.nlm.nih.gov/pmc/articles/PMC6044655/

A: Table 1 has been corrected in line 134

What you did in Table 1 with taking the lines is highly recommended for the other tables. Check out other papers. But again, it is not a major concern, but I believe that the MDPI editor will ask you for that…

A: Corrected all tables according to your suestion 

And, probably reviewer 2 asked for it, you do not need to place the units in each value. When you quote the table you can say that the values presented are in microvolts. Easier and faster (now that you had all that work, let it be).

A: You are absolutely right, we have left the unit only in the table caption, we hope this is clearer

In the figure, you wrote Alfa, while Alpha.

A: This has been corrected

I do not get the point of assessing Alpha in the frontal area… alpha is mainly in the occipital area, so for this work I do not understand. Furthermore, “The results of the present study suggest that high values were recorded for Alpha waves in certain leads, indicating a state of relaxation with closed eyes. This is in line with previous research in which it has been shown that the amplitude of Alpha waves usually increases during relaxation, which is characterized by a decrease in visual activity and a greater focus on one's own thoughts.” – You lack the reference(s) and you cannot say that your results are in line with the literature if the other studies do not show higher levels at the same sites as your study. This is a completely different story. If you find a study with the same results, on the same sites, then your results are in line with the literature.

A: References added, discussion improved by adding some content and references. Line 294-303

ALFA WAVES IN THE BRAIN They appear in the right hemisphere when solving mathematical tasks. Their emission in the left hemisphere increases when matching colours and shapes. If you are looking for non-obvious solutions and trying to tap into your creativity, Berger waves (a.k.a. Alpha waves). Appear in the temporal region, on the right side of the brain. Their frequency is 8-12 Hz. Their levels increase not only during creative tasks, but also in a relaxed state. They stimulate dreams and promote inspiration. Sharpened alpha waves in the occipital region signify deep relaxation. Alpha waves are also present in the frontal lobe. When present and repeated there, they can cause anxiety. This means that the patient is under tension. Perhaps facing a challenge. Berger waves that are too low can cause sleep problems, generate anxiety and restlessness. When Berger waves are, on the other hand, too high, we have difficulty concentrating and a feeling of lack of energy. Numerous studies report that the alpha rhythm is highly dependent on age and gender. As we age, the amplitude of alpha waves decreases, which can mean reduced creativity among older people and a decline in the ability to relax. ALFA WAVES FOR SCIENCE Alpha waves are also called Berger waves, after the German psychiatrist and creator of the EEG who described the types of waves and their rhythms. According to current research findings, these are the waves that create the best conditions for learning. Thanks to them, we have easier access to both hemispheres, which allows us to activate imagination and thinking, based on associations. What do alpha waves look like? The EEG allows us to observe their structure. These waves tend to be particularly prominent in the occipital leads. Their amplitude increases when we close our eyes. A decrease in the rhythm of alpha waves may indicate metabolic or dementia-related diseases. As Berger waves are responsible for activating the centres of emotion, intuition and creativity, they favour efficient and rapid learning and unconventional ideas

To improve your discussion about alpha, when you talk about relaxation, you can mention a recent article on how higher levels of the alpha band improve heart rate variability associated with lower levels of anxiety and stress (Domingos et al. 2021)*

A: Thank you for the hint we have added an excerpt along with the reference line 310-311

Overall, I do hope that the results you had were well cleaned of artifacts and others… in future works, I would recommend working with a neuroscientist with experience in Matlab and coding.

A:  Thank you, yes the results have been properly cleared. Thank you for your suggestion, we will certainly take your hints into account in future studies